# Inter-Oligomer Interaction Influence on Photoluminescence in Cis-Polyacetylene Semiconductor Materials

**DOI:** 10.3390/polym16131896

**Published:** 2024-07-02

**Authors:** Kamrun N. Keya, Yulun Han, Wenjie Xia, Dmitri Kilin

**Affiliations:** 1Department of Aerospace Engineering, Iowa State University, Ames, IA 50011, USA; knkeya@iastate.edu; 2Department of Chemistry and Biochemistry, North Dakota State University, Fargo, ND 58108, USA; yulun.han@ndsu.edu

**Keywords:** conjugated polymers, polyacetylene, ensembles of oligomers, photoluminescence, nonadiabatic couplings, nonradiative relaxation

## Abstract

Semiconducting conjugated polymers (CPs) are pivotal in advancing organic electronics, offering tunable properties for solar cells and field-effect transistors. Here, we carry out first-principle calculations to study individual cis-polyacetylene (cis-PA) oligomers and their ensembles. The ground electronic structures are obtained using density functional theory (DFT), and excited state dynamics are explored by computing nonadiabatic couplings (NACs) between electronic and nuclear degrees of freedom. We compute the nonradiative relaxation of charge carriers and photoluminescence (PL) using the Redfield theory. Our findings show that electrons relax faster than holes. The ensemble of oligomers shows faster relaxation compared to the single oligomer. The calculated PL spectra show features from both interband and intraband transitions. The ensemble shows broader line widths, redshift of transition energies, and lower intensities compared to the single oligomer. This comparative study suggests that the dispersion forces and orbital hybridizations between chains are the leading contributors to the variation in PL. It provides insights into the fundamental behaviors of CPs and the molecular-level understanding for the design of more efficient optoelectronic devices.

## 1. Introduction

In recent years, the study of conjugated polymers (CPs) has attracted significant interest, particularly because of their photoluminescent characteristics. The unique arrangement, featuring a sequence of alternating single and double bonds, is a critical factor contributing to the distinct properties of CPs [1,2]. These conjugations enable the electrons to delocalize through the polymeric chain. As a result of their properties, CPs have been used in many important applications such as organic photovoltaic [3], organic thermo-electrics [4], supercapacitors [5], electrochromic [6], biosensing [7], photocatalysis [8], optoelectronic device technology [9,10], such as light emitting diodes (LEDs) [2,11], transistors (thin-film transistors, TFTs) [11], and solar cells [9,11]. Currently, these compounds are among the most promising materials, due to their distinctive characteristics, such as processability, affordability, and thermal stability in film formation [12,13]. A considerable amount of experimental and theoretical work has been dedicated to elucidating the electronic and optical characteristics of CPs [14].

CPs often have structures that are amorphous or semi-crystalline [15]. They exhibit short-range order without long-range regularity, which complicates their solubility and results in polydispersity. This makes it challenging for computational models to accurately represent these polymers with a variety of sizes and shapes. Consequently, the properties of shorter oligomeric chains have been studied, from which one can infer the characteristics of much longer polymeric chains. However, accurate electronic properties of polymers cannot simply be extrapolated from oligomers, as revealed by Hutchison et al. [16]. Studies have shown that, through meticulous chemical design involving the strategic placement of fluorine atoms within the polymer’s backbone, aqueous CP nanoparticles exhibit elevated photoluminescence (PL) quantum yields, particularly in the far-red and near-infrared (NIR) regions [17]. The functionalization of CPs with flexible side chains, including alkyl groups, leads to the enhancement of solubility and device performance. These side chains do not directly participate in charge transport, but they can influence the overall device performance by affecting the molecular packing and physical robustness of the films [18].

Among the diverse CPs, polyacetylene (PA) is distinguished by its prototypical role and noteworthy photoelectric qualities [19]. Monosubstituted PA exhibits a regular helical structure that can be fine-tuned and polymerized under non-stringent conditions. These attributes allow for precise control over their luminescence properties, making them suitable for various PL applications [20,21]. The singular dimensionality of CPs facilitates soliton formation, which impacts the charge separation and transport within these materials [22,23]. Photogenerated excitations in trans-PA tend to relax into soliton states, significantly reducing PL in the visible range while primarily confining it to the infrared range. This characteristic, stemming from the unique topological and electronic structures of trans-PA, inherently limits its effectiveness in applications requiring visible-range PL. In contrast, cis-PA does not exhibit this soliton-induced reduction in PL, making it suitable for broader applications in optoelectronic devices, where efficient and strong light emission is essential [22,24].

The application of density functional theory (DFT) is essential for studying the electronic structures of semiconductors, providing a comprehensive understanding of their ground states and mechanisms involved in charge transfer at material interfaces [25]. The computational methodologies divide the description of charge transfer into two distinct segments: the initial structural modeling and electronic state analysis, and the subsequent categorization and dynamic assessment of excited states. The interplay between dissipative quantum dynamics and molecular trajectories, informed by DFT and refined through approaches such as surface hopping [26,27,28,29,30,31] and Redfield [32,33,34], remains a rich area of research to unravel the complexities of charge-transfer excitations and their temporal evolution.

In previous work [35], we focused on the ground state properties of ensembles of cis-PA oligomers. We have also studied the nonradiative relaxation of charge carriers and PL of individual cis-PA oligomers [36]. In this study, we examine the excited state dynamics of ensembles of oligomers, comparing the results with those of individual oligomers. Specifically, the ground electronic structures are obtained from DFT calculations. We then compute nonadiabatic couplings (NACs) between electronic and nuclear degrees of freedom along adiabatic molecular dynamic (MD) trajectories. The excited state dynamics is computed using the reduced density matrix (RDM) [37,38,39] formalism within the Redfield theory. This method provides a deeper understanding of the complex behavior of charge carriers, enabling us to map out the possible relaxation processes for both individual and ensembles of oligomers.

## 2. Methods

Our methodology consists of two components: the assessment of the ground state electronic structures using DFT and the examination of excited state dynamic processes through calculations of NACs based on the Redfield theory. This methodology has been applied to various molecular [40,41] and periodic systems [42,43,44,45,46] by some of us to investigate photoinduced processes. In each section, we compute several observables, which facilitate the investigation of charge carrier relaxation of individual cis-PA oligomers and their ensembles.

### 2.1. Ground-State Calculations

We begin by defining the atomic model through the initial ionic positions ${\vec{R}}_{I}$. The electronic structure is then determined by solving the self-consistent field equations of DFT. The core of this methodology is the fictitious one-electron Kohn–Sham (KS) framework [47,48]. The KS formalism is integral to our DFT calculations, enabling us to refine the estimates of orbital energies.
(1)$$\left(-\frac{\hbar^{2}}{2m}\nabla^{2}+v\left[\;\left\{{\vec{R}}_{I}\right\},\;\vec{r,}\;\rho \left(\vec{r}\right)\right]\right)\varphi_{i}^{KS}\left(\;\left\{{\vec{R}}_{I}\right\},\;\vec{r}\right)=\varepsilon_{i}{\left(\;\left\{{\vec{R}}_{I}\right\}\right)\varphi }_{i}^{KS}\left(\;\left\{{\vec{R}}_{I}\right\},\vec{r}\right)$$

The wave functions $\varphi_{i}^{KS}(r),$ correspond to the KS orbitals, and $\varepsilon_{i}$ are their respective eigen energies. In the Appendix A describe the total density of electrons and states for all orbitals.

The absorption spectrum is defined as:(2)$$a\left(\varepsilon \right)=\sum_{ij}^{\;}f_{ij}\delta (\varepsilon -\Delta \varepsilon_{ij})$$
where $f_{ij}$ represents the oscillator strength, which quantifies the likelihood of transition from the initial state i to the final state j, and $\Delta \varepsilon_{ij}$ is the energy difference between these states. $f_{ij}$ is calculated as
(3)$$f_{ij}={\left|{\vec{D}}_{ij}\right|}^{2}\frac{4\pi m_{e}v_{ij}}{3\hbar e^{2}}$$
where $m_{e}$ represents the electron mass, ℏ the reduced Planck’s constant, and $v_{ij}$ the resonant frequency associated with the transition from initial state i to final state j. The transition dipole moment ${\vec{D}}_{ij}$ is a vector quantity that encapsulates the electric dipole moment matrix element associated with the transition and is calculated as:(4)$${\vec{D}}_{ij}=e\int_{\;}^{\;}\varphi_{i}^{KS^{*}}\vec{r}\varphi_{j}^{KS}d\vec{r}$$This integral combines, the cumulative effect of matching metween spatial distribution of electron in the initial and final state on the transition dipole moment.

KS orbitals are visualized as three-dimensional iso-surfaces, representing the spatial distribution of electron density or through one-dimensional density profiles along a specified axis, given as:(5)$${\rho_{i}(z)=\iint dxdy\left|\varphi_{i}^{KS}\left(x,y,z\right)\right|}^{2}$$
(6)$${\rho_{i}(y)=\iint dxdz\left|\varphi_{i}^{KS}\left(x,y,z\right)\right|}^{2}$$These equations provide a detailed view of the electron density distribution across different spatial dimensions, essential for interpreting the electronic behavior of the system.

### 2.2. Excited-State Calculation

The dissipative transitions are calculated within the framework of adiabatic MD trajectories. The details are available in previous work [34]. Briefly, the positions of ions $\left\{{\vec{R}}_{I}\left(t=0\right)\right\}$ and velocities at the initial time $\left\{\frac{d}{dt}{\vec{R}}_{I}\left(t=0\right)\right\}$ are factored into the MD simulations. This initial condition enables the calculation of NACs:(7)$$V_{ij}\left(t\right)=\frac{-i\hbar }{\Delta t}\int_{\;}^{\;}d\vec{r}\;{\varphi_{i}^{KS}}^{*}\left(\;\left\{{\vec{R}}_{I}\left(t\right)\right\},\;\vec{r}\right)\varphi_{j}^{KS}\left(\;\left\{{\vec{R}}_{I}\left(t+\Delta t\right)\right\},\;\vec{r}\right)+h.c.\;\;$$

The autocorrelation function of these couplings is anticipated to peak initially and decline rapidly thereafter, contributing to the components of the Redfield tensor:(8)$$M_{ijkl}\left(\tau \right)=\frac{1}{T}\int_{0}^{T}dtV_{ij}\left(t+\tau \right)V_{kl}\left(t\right)$$

A Fourier transform of the coupling autocorrelation function provides the partial component $\Gamma^{\pm }$ as follows:(9)$$\Gamma_{ljik}^{+}=\int_{\;}^{\;}{d\tau \;M}_{ljik}\left(\tau \right)exp\;\left(-i\omega_{ik}\tau \right)$$
(10)$$\Gamma_{ljik}^{-}=\int_{\;}^{\;}{d\tau \;M}_{ljik}\left(\tau \right)exp\;(-i\omega_{lj}\tau )$$

Equations (8)–(10) facilitate the expression of the autocorrelation function’s partial components. With these, one can obtain the Redfield tensor, which is pivotal in managing the dynamics of the density matrix:(11)$$R_{ijkl}=\Gamma_{ljik}^{+}+\Gamma_{ljik}^{-}-\delta_{jl}\sum_{m}^{\;}\Gamma_{immk}^{+}-\delta_{ik}\sum_{m}^{\;}\Gamma_{jmml}^{-}$$
(12)$${\left(\frac{d\rho_{jk}}{dt}\right)}_{diss}=\sum_{lm}^{\;}R_{jklm}\rho_{lm}$$

The initial excitation by a photon ${\hbar \Omega }_{AB}$ signifies the transition energy difference between orbitals A and B. At time t=0, the excitation energy is characterized within the framework of the density matrix:(13)$$\rho_{ij}\left(0\right)=\delta_{ij}\left(f_{i}-\delta_{iA}+\delta_{jB}\right)$$
where $f_{i}$ refers to the equilibrium thermal population of the ith orbital based on the Fermi–Dirac distribution. To determine how the electronic state evolves over time, one can employ the equation of motion, written as:(14)$${\dot{\rho }}_{ij}=-\frac{i}{\hbar }\sum_{k}^{\;}(F_{ik}\rho_{kj}-\rho_{jk}F_{ki})+{\left(\frac{{d\rho }_{ij}}{dt}\right)}_{diss}$$

We can understand the electronic transitions between orbitals by studying the thermal fluctuations of the ions of the system. The computational solution of Equation (14) furnishes the temporal evolution of the density matrix elements $\rho_{kj}(t)$, while $F_{ik}$ denotes the Fock matrix elements.

Dissipation rates are derived from Equation (11). Essential to this are the system’s diagonal elements $\rho_{jj}(t)$ illuminated by the time-varying occupations of the KS orbitals. With these data, we can deduce the charge density distribution, the energy dissipation rate, and the charge transfer rate within the system. Consequently, the charge distribution of the system can be conceptualized as:(15)$$n^{\prime \prime }\left(\varepsilon ,t\right)=\sum_{i}^{\;}\rho_{ii}\left(t\right)\delta (\varepsilon -\varepsilon_{i})\;$$With Appendix A on the relaxation for electrons and holes, we proceed to estimate the relaxation rates of charge carriers towards the band edges. The process requires the transformation of the energy expectation value into a dimensionless form, which is then modeled with a single-exponential decay to obtain the rate constant:(16)$$k_{e}={\left\{\tau^{e}\right\}}^{-1}=\int_{0}^{\infty }{\left\{\langle E_{e}\rangle \left(t\right)dt\right\}}^{-1}$$
(17)$$\rho \left(\vec{r},t\right)=\sum_{ij}^{\;}\rho_{ij}\varphi_{i}^{*}\left(\vec{r}\right)\varphi_{j}\left(\vec{r}\right)$$
(18)$$∆\rho \left(\vec{r},t\right)=\rho \left(\vec{r},t\right)-\rho^{eq}\left(\vec{r}\right)$$
(19)$$∆n\left(z,t\right)=\int_{\;}^{\;}dx\int_{\;}^{\;}dy∆\rho \left(\vec{r},t\right)$$

The populations of charge carriers and distribution energies can be described by excitation energy:(20)$$P\left(\varepsilon ,t\right)=\sum_{j\leq HO}^{\;}\sum_{i\geq LU}^{\;}\rho_{ii}\left(t\right)\cdot \rho_{jj}\left(t\right)\delta \left(\varepsilon_{i}^{E}-\varepsilon_{j}^{H}-\varepsilon \right)$$
where $\varepsilon_{i}^{E}$ and $\varepsilon_{j}^{H}$ represent the energies of electron and hole orbitals, respectively. The time-integrated emission $E\left(\hbar \omega \right)$ can be calculated based on the time-resolved emission $E\left(\hbar \omega ,\;t\right)$, which involves the product of the oscillator strength $f_{ij}$ and the delta function δ of energy difference and the electronic populations $\rho_{jj}\left(t\right)$ and $\rho_{ii}\left(t\right)$:(21)$$E\left(\hbar \omega ,\;t\right)=\sum_{j>i}^{\;}f_{ij}\delta \left(\hbar \omega -\hbar \omega_{ij}\right)\left\{\rho_{jj}\left(t\right)-\rho_{ii}(t)\right\}$$
(22)$$E\left(\hbar \omega \right)=\frac{1}{T}\int_{0}^{T}E\left(\hbar \omega ,\;t\right)dt$$The emission occurs between the pairs of orbitals when the inverse population condition meets the $\rho_{jj}>\rho_{ii}$, $\varepsilon_{j}>\varepsilon_{i}$ criterion.

The inverse participation ratio (IPR) [49] can be computed from normalized displacement vectors and is expressed as:(23)$$I\left(\psi_{j}\right)=\frac{N\sum_{i=1}^{N}a_{i}^{4}}{\sum_{i=1}^{N}a_{i}^{j2}}$$
where $a_{i}^{j}$ are the components of the *j*th KS orbital projected onto the basis states. $\psi_{j}={\sum_{i=1}^{N}a_{i}^{j}\varphi }_{i}$ is the *j*th KS orbital, $\varphi_{i}$ is the orthogonal basis, and N is the number of basis functions. The IPR serves as a metric for the degree of localization within electronic states. Highly localized states are indicated by an IPR approaching 1, whereas completely delocalized states, uniformly spread over N atoms, are reflected by an IPR of 1/N.

### 2.3. Computational Details

DFT calculations are performed using the Vienna Ab initio Simulation Package (VASP) [50,51,52,53]. The exchange–correlation effects are accounted for by the Heyd–Scuseria–Ernzerhof (HSE06) [54] hybrid functional. This study is conducted on two distinct classes of cis-PA, with their atomistic structures displayed in Figure 1. Panel (a) illustrates the optimized geometry of a single oligomer, with a molecular formula of C_32_H_36_, placed in to a periodic box of the size of 12.1 Å×12.8 Å×43.8 Å, with vacuum between periodic images. Meanwhile, panel (b) illustrates an ensemble of oligomers, places in a periodic box of the size 36.0 Å×5.7 Å×19.6 Å, depicting the collective behavior and interactions within a condensed phase. There are six oligomers with a net molecular formula of C_184_H_208_. In the ensemble of oligomers, we reduce the number of C and H from the 2nd and 5th oligomer to host dopants, which will be explored in future studies.

Before conducting dynamic analyses, all models are relaxed to optimal geometries in the ground state. For dynamic processes, the models are preheated and maintained using a thermostat, followed by MD simulations at 300 K. All simulations are performed at the gamma point. It is important to mention that our nonadiabatic analyses average results across a microcanonical ensemble (NVE ensemble), where 1 ps MD trajectories with a time step of 1 fs are sampled to capture a broad range of possible states that the actual system may encounter. In our approach, we integrate both electronic and nuclear degrees of freedom through adiabatic MD simulations and compute NACs among adiabatic states via an on-the-fly technique [55].

For the ensemble model, oligomers are placed at a close distance to balance the pressure between unit cells and achieve a realistic density. The optimal distance depends on dispersion corrections implemented by the D3 method [56]. The distance between the nearest (5th and 6th) chains is 3.18 Å. Note that the periodic boundary conditions are applied in both models. We explore the computational efficiency of forming periodic boxes filled with both shorter and longer oligomers, which could potentially be extended in future work. Changing the size of the box and practically thermal annealing would likely result in additional defects such as twists and kinks in individual oligomers. The current work corresponds to the low-temperature crystallization limit and can be considered as a reference point.

## 3. Results

Figure 2a describes the energy distributions of the frontier orbitals for the single oligomer. We utilize dual notations for these orbitals: n=1, 2, 3 for the unoccupied orbitals and $n^{\prime }=1,\;2,\;3$ for occupied ones. The energy of the lowest unoccupied molecular orbital, i.e., LU (n=1), is at −3.0 eV and that of the highest occupied molecular orbital, i.e., HO $\left(n^{\prime }=1\right)$, is at −4.2 eV. When examining higher n values, there is an increase in the energy levels of the unoccupied orbitals. This notation is according to the quantum confinement paradigm. A way to count orbitals of electrons upward from the conduction band minimum (CBM) and holes downward from the valence band maximum (VBM) is an efficient way to establish an analogy with momentum dispersion in direct gap infinite polymers; $\varepsilon_{CBM}\left(p\right)=\varepsilon_{CBM}\left(0\right)+\frac{p^{2}}{2{m^{*}}_{e}\;}$; $\varepsilon_{VBM}\left(p\right)=\varepsilon_{VBM}\left(0\right)-\frac{p^{2}}{2{m^{*}}_{h}\;}$. Note the negative size of dispersion for holes. For finite size oligomers, both e,h follow the trend of a particle in a box, reflected in the special pattern of subgaps. These background concepts provide a foundation for a more organized analysis of electronic structures.

Our findings from the single oligomer model indicate an energy gap $E_{gap}$ of 1.2 eV between the HO and LU orbitals. Furthermore, Figure 2b illustrates that the gap between HO and LU is more pronounced than the subgaps noted between HO and HO−1, as well as LU and LU+1 orbitals for the single oligomer. The symbols A, B, C, D, and M are used for the single oligomer, and A′ and M′ are used for the ensemble of oligomers. A−D represent transitions from HO to LU (A), HO−1 to LU (B), HO to LU+1 (C), and HO−1 to LU+1 (D). A′ shows the transition from HO to LU ($E_{gap}\;of\;0.8\;eV$). A−D and A′ correspond to interband transitions, whereas M and M′ correspond to intraband transitions that may arise in PL.

Although the ensemble of cis-PA oligomers follows the pattern of a particle in a box, there is a slight difference from the single oligomer. There are π−π interactions for the ensemble of oligomers. Due to inter-oligomer interactions, the degenerate states of individual oligomers are lifted in the ensemble. The spatial proximity of the oligomers induces the interaction between their individual eigenstates. Generally, an interaction between any two quantum levels results in an increase in their energy offset. If the offset is zero, indicating degeneracy, the interaction lifts this degeneracy, making the offset nonzero. Compared to a single oligomer, an ensemble of oligomers has a greater chance of hybridization between the electronic states.

In our earlier work [36], we did show that TDDFT calculations with a hybrid functional provide gap values comparable to experimental results. By comparing DFT and TDDFT data, one may extract a systematic correction. For a single oligomer, $E_{gap}\left(DFT\right)=1.2\;\mathrm{eV}$ and $E_{gap}\left(TDDFT\right)=1.9\;\mathrm{eV}$. The difference is $∆={E_{gap}(TDDFT)-E}_{gap}\left(DFT\right)=0.7\;\mathrm{eV}$. Thus, each transition energy in our calculations may be corrected as follows: E(i,j,corrected)=E(i,j, computed)+∆. An alternative way is to scale the energies by a factor of R, where $R=E_{gap}(TDDFT)\;/\;E_{gap}\left(DFT\right)=1.6$. In this approach, all transition energies in our calculations may be corrected as follows: E(i,j,corrected)=E(i, j, computed)∗R.

Figure 2c illustrates the spatial iso-contours of the frontier orbitals for a single oligomer of cis-PA. Each orbital demonstrates two properties: first, a repeating pattern in the vicinity of each carbon atom, and second, a slowly changing envelope function that modulates the density near each atom as a function of position along the oligomer. Notably, the LU orbital demonstrates the maximum amount of the charge density peaks within its central domain, whereas the edges display reduced densities. In analogy to the particle in a box model, the maximum charge density is located centrally in the box, with diminished densities at the edges. For n=2, the charge density’s peak shifts from its initial position, indicative of the eigenstate’s index increment. This results in the orbital splitting into two segments, with the highest charge density concentrated in the central regions and negligible density at the edges, as depicted for the ensemble of cis-PA oligomers in Appendix A. The ensemble of oligomers exhibits a charge distribution pattern akin to the particle in a box model.

The calculated absorption spectra for the single and ensemble of oligomers are reported in Figure 3. The peaks are labeled in the order of ascending transition energy. The initial peak A′ and A are observed at 0.8 eV for the ensemble and at 1.2 eV for the individual oligomer, respectively, indicative of transitions from HO to LU. The other peaks in the spectra are representative of various electronic transitions occurring among different orbitals. Peaks for an ensemble of oligomers are redshifted, broader, and of lower intensity compared to peaks for a single oligomer. The broadening is due to inter-oligomer interactions. Table 1 provides representative transitions based on peak labels from Figure 3, which are used as initial conditions for excited state dynamics.

In the next step, we explore nonradiative transitions between electronic states. The RDM approach and Redfield formalism provide components of the Redfield tensor after processing the autocorrelation function of NACs. The Redfield tensors provide the electronic level-to-level transition rates. Figure 4a–d demonstrate the autocorrelation function of NACs and the elements of the Redfield tensor, respectively, for the single and ensemble of oligomers. Using the autocorrelation function, we can explain the average nonadiabatic interaction in the electron–lattice interaction. This autocorrelation function is the correlation function, which provides information about the intensity of dissipative electronic transitions for the given indices i,j,k, l with several amplitudes. Interestingly, for any tested combination of indices, the autocorrelation function decays abruptly within 5 fs for the single oligomer and within 2 fs for the ensemble. The ensemble decays at a faster rate than the single oligomer, since inter-chain interactions affect electronic transitions.

Dissipative excited state dynamics are provided in Figure 5. There are two distinct sectors, each marked by a different color. The yellow sections signify an electron charge surplus (∆n>0), while the blue sections denote a shortage in hole charge (∆n<0), compared to the neutral charge distribution $\left(∆n=0\right)$. Figure 5a visualizes the nonradiative relaxation of the photoexcited electrons and holes for a single oligomer, which can take ~15 ps and ~150 ps, respectively. The energy dissipation dynamics are tracked through the expected values of energy for both electrons and holes. Between roughly ~10 ps and ~15 ps, one observes the electron relaxation from LU+1 to LU. Following a period of around 150 ps, there is an observed energy shift for the holes. This increase marks the migration of the hole population from HO−1 to HO.

Figure 5b delves into the progression of the non-equilibrium charge density distribution along the z-axis. The moments when electrons and holes achieve maximum charge density transfer are marked by vertical lines for electrons $\tau_{e}$ and for holes $\tau_{h}$, respectively. The change in spatial distribution of charge can be interpreted in terms of the pattern of a particle in a box, observed for progressions of occupied and unoccupied orbitals. The central and edge regions of the box lack yellow lines in the range of 0~15 ps due to the specific pattern of the HO−1 and LU+1 orbitals as depicted in Figure 2. After 15 ps, the yellow lines have already condensed in the center, and the blue areas are still available. This corresponds to LU with maximal density in the central area. We observe that at a time between 15 ps and 150 ps, the electrons and holes stay in different spatial areas of the single oligomer. At ~150 ps, the blue lines also condense in the center of the oligomer as a signature of HO getting populated.

Figure 5d,e illustrates the dynamic results for the ensemble of oligomers. The initial photoexcitation condition is HO−6 → LU+4. At the initial time, the localization of holes in oligomers 3 and 4 is evident, while electrons are primarily localized in oligomers 1 and 6. Then, one sees the hot charge carriers undertaking stepwise changes until the electrons and holes reach frontier orbitals, and in the process dissipating excess energy transforms to heat. Along the excited state trajectories, the ensemble of oligomers experiences the subsequent population of several intermediate electronic states. At 1~5 ps, electrons and holes stay in different areas of the space. Specifically, electrons stay at oligomers 1 and 6, whereas holes stay at oligomers 2, 3, and 4. Finally, after ~5 ps, electrons are at oligomers 5 and 6, whereas holes are at oligomers 1, 2, and 3. Appendix A depicts HO−1, HO, LU, and LU+1 orbitals, which clearly shows the localization pattern.

Table 2 and Table 3 summarize the charge carrier relaxation rates for the single oligomer and ensemble of oligomers under different initial photoexcitation conditions. It is found that electrons relax faster than holes in both single and ensemble of oligomers. This is due to smaller subgaps and more pronounced phonon interactions for electrons than holes. Additionally, the ensemble of oligomers shows faster relaxation compared to the single oligomer. This originates from the higher density of states and a larger number of relaxation pathways in the ensemble compared to the single oligomer. In Figure 6, we explore the dependence of relaxations rates of charge carriers on dissipated energies. For both models, one finds the decrease in relaxation rates for electrons and holes with increasing dissipated energies in accordance with the so-called energy gap law [57].

Figure 7 depicts the dynamics of radiative relaxation for the single oligomer and ensemble of oligomers. The initial conditions are the same as in Figure 5. Panels (a, d) show the dynamics of excitation energy dissipation. In these spectra, one finds features A–C for the single oligomer and features A′–C′ for the ensemble of oligomers. These features are due to the cascading down of excitation energy from the initial excitation to the lowest excitation. Panels (b, e) show the time-resolved emission spectra. For the ensemble, one finds several emission features with vanishing intensities. Peaks A′ and C′ are more pronounced than peak B′. When the population cascades, it may visit the dark states. The main features are peaks M and M′, which correspond to intraband emission features below the band gap. These intraband transitions appear in the emission spectra but not in the absorption spectra. Panels (c, f) show the time-integrated emission spectra. Peaks in these spectra are consistent with peaks from the time-resolved emission spectra. However, some emission characteristics such as B′, which are quite faint and almost imperceptible in time-resolved emission spectra, become apparent in the time-integrated emission spectra. The onset of these spectral features occurs later in the dynamic process. This prolonged activity contributes significantly to the time-integrated emission spectra.

The emission spectra of both models demonstrate both interband and intraband emission. Peaks C and C′ corresponding to the initial excitations will vanish for lower energy excitations. At sufficiently high energy excitations, these peaks will be visible. The current initial conditions are intentionally chosen to simplify analysis. The ensemble of oligomers shows three effects: broader line widths, redshift of transition energies, and lower intensities compared to the single oligomer.

Figure 8 illustrates the dynamics and spatial distribution of electrons and holes for the single oligomer and ensemble of oligomers. It also showcases the varying nature of molecular orbitals, which can be observed as either localized or delocalized. These patterns are directly associated with the spatial distribution of electrons and holes of each system. The initial conditions are the same as in Figure 5. For the single oligomer, the delocalization patterns for both electrons and holes are visible at the initial time. After relaxation, electrons are more localized and holes are more delocalized. Comparing electrons with holes throughout the relaxation pathway, electrons remain more localized than holes. For the ensemble, electrons are delocalized over four oligomers, and holes are localized over two oligomers at the initial time. After relaxation, both electrons and holes are more localized. Comparing electrons with holes throughout the relaxation pathway, electrons are generally less localized than holes, except in the intermediate states, where both electrons and holes show greater delocalized patterns.

## 4. Discussion

A systematic study on the dependence of chain length is known in the literature and is ongoing as a continuation of this work. An increase in chain length is expected to provide a mild redshift of transition energies. Our exploration into models with shorter-length oligomers suggests a decrease in computed conductivity. This decrease is attributed to quantum confinement effects in shorter oligomers, which increase band gaps and subgaps.

In the context of CPs, oligomers demonstrate the formation of bound excitons of finite sizes, primarily due to electron–hole interactions. Brey and Burghardt [58] elucidate a coherent transient localization mechanism for exciton transport in regioregular poly(3-hexylthiophene), highlighting the interplay of exciton polarons and interchain vibrational modes. In shorter oligomers, the confinement is more pronounced, affecting the electronic properties significantly. In contrast, longer oligomers exhibit transition energies that stabilize and are defined predominantly by the size of the exciton and polaron rather than by further increases in length. This behavior aligns with observations in the literature, suggesting a general trend for exciton behavior in CPs. For instance, Kraner et al. [59] find that exciton sizes increase uniformly with molecular π-system sizes ranging from 10 to 40 Å. Beyond this range, no significant increase in exciton size is observed. These findings guide our decision to utilize shorter oligomers.

We focus on the influences and consequences of inter-chain interaction on nonradiative relaxation and spectral line widths. The fundamental photophysical properties such as charge carrier dynamics and PL behavior are inherent to cis-PA. These properties demonstrate consistency across the different molecular weights we have studied, suggesting that the trends we observe could extend to cis-PA products with broader molecular weights. As soon as the chain length exceeds the size of the polaron, its influence on such observables becomes negligible. There is ongoing research to confirm this hypothesis. Adding additional defects, such as kinks, non-planar structures, and twists, will provide additional effects. This approach aligns with the comprehensive design strategies outlined in the literature, providing a foundation for our continued research into developing more efficient and application-specific CPs. In the current study, dispersion forces and orbital hybridizations between chains are the major effect.

The incorporation of methyl groups as terminal substituents in our oligomer configurations offers significant benefits. Methyl groups enhance molecular stability by providing steric protection, which reduces the susceptibility of the polymers to oxidative degradation. This protection is crucial for maintaining the structural integrity of the oligomers during experimental procedures and analyses. Additionally, these methyl groups significantly influence the electronic structures of the oligomers. They adjust the electron density along the polymer backbone, thereby affecting the band gap and electronic transitions crucial for the PL properties under study. The strategic inclusion of methyl groups modifies the energy states of HO and LU, enhancing the optical properties of the oligomers. Therefore, the choice to use methyl groups instead of hydrogen atoms is not merely a structural decision but a strategic one aimed at optimizing the optoelectronic properties of the oligomers.

The current investigation is designed to deepen the theoretical understanding and enhance the accuracy of computational models within a defined theoretical framework. While experimental validation, partially provided in ref. [60], remains critical for the broader application of these computational study, the present study is limited to comparative analyses between different computational models. The computation-to-experiment comparison, while highly valuable, falls outside the scope of the current study. We expect that adding the scissor operator or scaling correction would lead to accurate transition energies and bring the spectra in computational results closer to experimental values.

## 5. Conclusions

We carry out first-principles calculations to systematically compare static properties and photoexcited dynamics between individual cis-PA oligomers and their ensembles. The ground electronic structures are obtained using DFT. The excited state dynamics are explored by computing NACs between electronic and nuclear degrees of freedom. The nonradiative relaxation of charge carriers and PL are computed using the RDM formalism within the Redfield theory. Our findings quantify relaxation timescales, enhancing the design and efficiency of photovoltaic materials. Specifically, it is found that electrons relax faster than holes in both the single and ensemble of oligomers. This is due to smaller subgaps and more pronounced phonon interactions for electrons than holes. Additionally, the ensemble of oligomers shows faster relaxation compared to the single oligomer. This originates from the higher density of states and a larger number of relaxation pathways in the ensemble of oligomers compared to the single oligomer.

PL spectra of both models demonstrate interband and intraband emissions. The ensemble of oligomers shows three effects: broader line widths, redshift of transition energies, and lower intensities compared to the single oligomer. Our comparative study suggests that the dispersion forces and orbital hybridizations between chains are the leading contributors to the variation in PL. We thus expect that the functionalization of CPs by side chains will mitigate inter-oligomer interactions. This approach is anticipated to enhance the intensity and stability of PL. This work provides a detailed analysis of the optoelectronic properties of cis-PA, which could be used for improving nanostructured semiconductor materials for photovoltaic and film-forming conductive polymer processing.

## Figures and Tables

**Figure 1 polymers-16-01896-f001:**
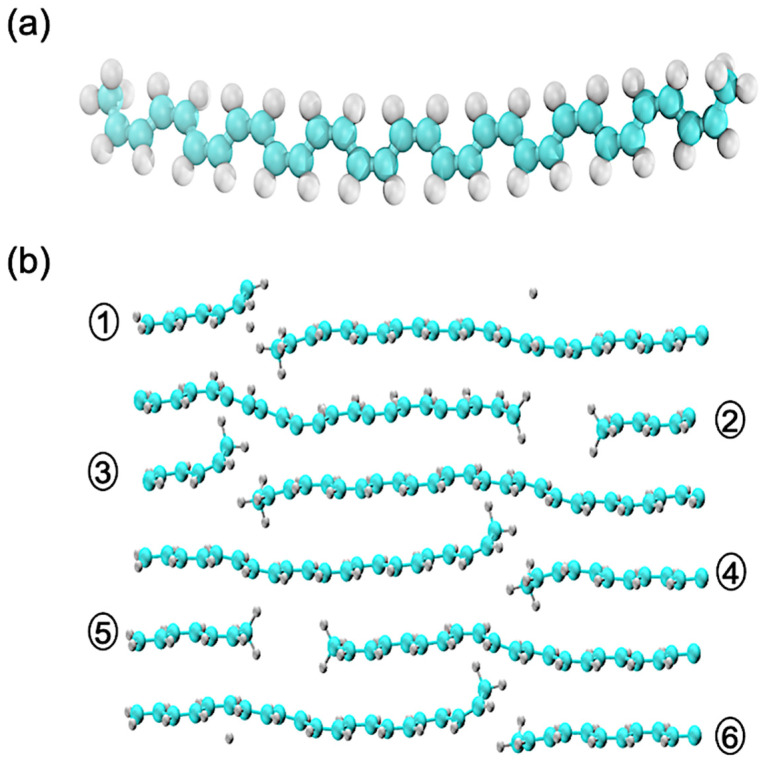
Optimized geometry of individual and ensemble of cis-PA. (**a**) Single oligomer aligned in the yz plane with a molecular formula of C_32_H_36_. (**b**) Six oligomers with a molecular formula of C_184_H_208_. It is important to note that the periodic box encapsulating the oligomers is configured to be more compact than the extent of an individual oligomeric chain. These oligomers are numbered from 1 to 6 in ascending order of their position along the z-coordinate. This alignment permits the entire ensemble to be replicated seamlessly along the z-axis.

**Figure 2 polymers-16-01896-f002:**
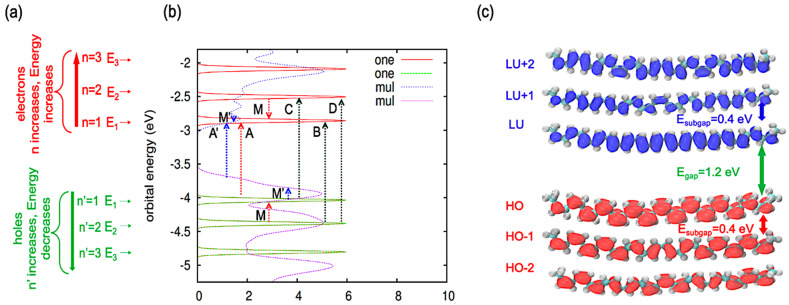
Comparative electronic structures of cis-PA oligomers. (**a**) Orbital energy levels for a single oligomer, with energies corresponding to occupied (n′) and unoccupied (n) KS orbitals, indicative of particle-in-a-box behavior. Transitions are marked by paired indices. (**b**) DOS for the single oligomer and ensemble of oligomers. A, B, C, D, M symbols are used for the single oligomer and A′ and M′ are used for the ensemble of oligomers. Symbols A−D and A′ correspond to interband transitions, whereas M and M′ correspond to intraband transitions that may arise in PL. The term ‘one’ pertains to the single oligomer, while ‘mul’ denotes the ensemble. (**c**) Charge densities for the orbitals from HO−2 to LU+2 for the single oligomer are depicted.

**Figure 3 polymers-16-01896-f003:**
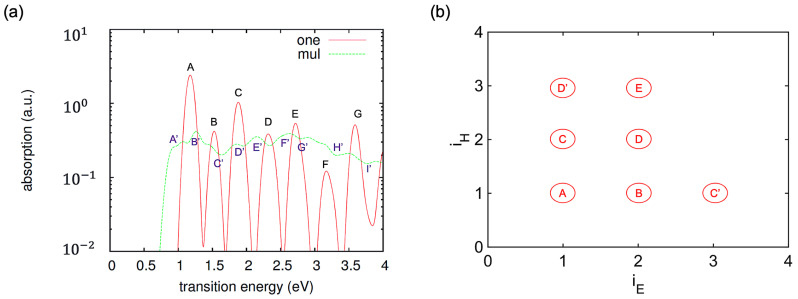
(**a**) Calculated absorption spectra for the single oligomer and ensemble of oligomers. Label “one” refers to the single oligomer and label “mul” refers to the ensemble of oligomers. (**b**) Schematic diagram illustrating the composition of absorption peaks on the basis of transitions between pairs of independent orbitals for both single and ensemble of cis-PA.

**Figure 4 polymers-16-01896-f004:**
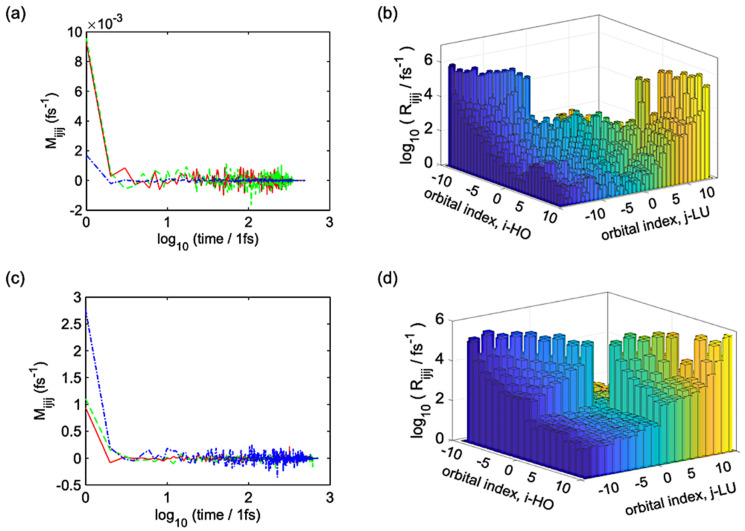
(**a**,**c**) Autocorrelation function and (**b**,**d**) Redfield tensor of the single and ensemble of oligomers. (**a**,**b**) correspond to the single oligomer and (**c**,**d**) correspond to the ensemble. Autocorrelation functions correspond to orbital pairs: HO−1 to HO (green), HO to LU (blue), and LU to LU+1 (red).

**Figure 5 polymers-16-01896-f005:**
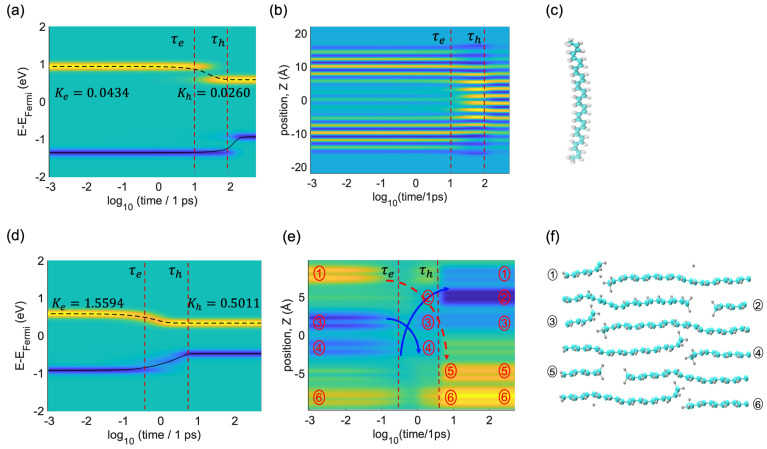
Relaxation dynamics of the (**a**–**c**) single and (**d**–**f**) ensemble of oligomers. The initial conditions are HO−1→LU+1 and HO−6→LU+4 for the single and ensemble, respectively. (**a**,**d**) display dynamics of distribution of charge carriers in energy. (**b**,**e**) display dynamics of the spatial distribution of charge carriers. The iso-contours of charge variation track the temporal progression of electron (in yellow) and hole (in blue) populations, with the turquoise shade indicating the baseline ground state distribution. Vertical lines labeled as $\tau_{e}$ and $\tau_{h}$ indicate the time of the population transfer between the electron and hole orbitals. (**c**,**f**) display atomistic models, where turquoise color spheres represent carbon and white color spheres represent hydrogen. Roman numbers in circles provided in panels (e) and (f) lable spatial positions of individual oligomers.

**Figure 6 polymers-16-01896-f006:**
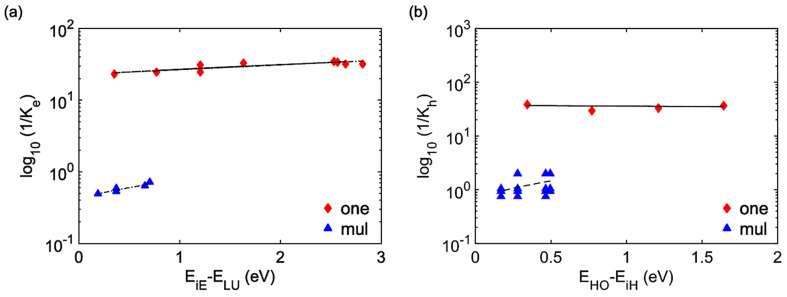
Dependence of relaxation rates of electrons and holes on dissipated energies. (**a**) corresponds to hot electrons. (**b**) corresponds to hot holes. Red color indicates the single oligomer (designated as ‘one’), and blue color indicates the ensemble of oligomers (designated as ‘mul’). $E_{iE}$ refers to the energy of the electrons, $E_{iH}$ refers to the energy of the holes, $E_{LU}$ indicates the energy of LU, and $E_{HO}$ indicates the energy of HO.

**Figure 7 polymers-16-01896-f007:**
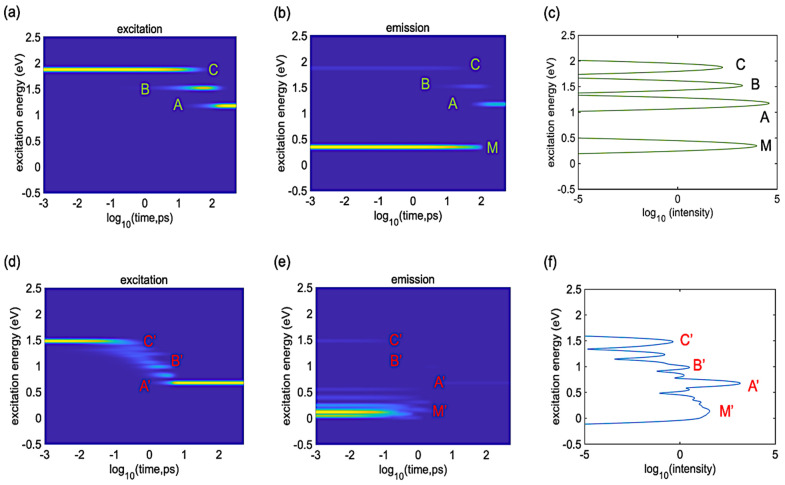
Computed radiative relaxation dynamics for the (**a**–**c**) single oligomer and (**d**–**f**) ensemble of oligomers. The initial conditions are HO−1→LU+1 and HO−6→LU+4 for the single and ensemble, respectively. (**a**,**d**) show the time-dependent energy dissipation of excitons. (**b**,**e**) show the time-resolved emission spectra. The color gradient from yellow to navy indicates high to zero population levels. (**c**,**f**) show the time-integrated emission spectra. Peaks A–C and A′–C′ correspond to interband transitions, whereas peaks M and M′ correspond to intraband transitions.

**Figure 8 polymers-16-01896-f008:**
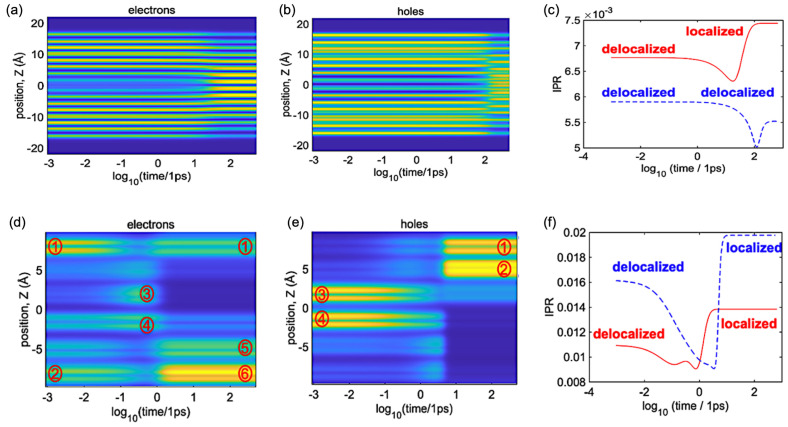
Dynamics of spatial distribution for the (**a**–**c**) single oligomer (**d**–**f**) and ensemble of oligomers. The initial conditions are HO−1→LU+1 and HO−6→LU+4 for the single and ensemble, respectively. (**a**,**d**) correspond to electrons. (**b**,**e**) correspond to holes. (**c**,**f**) summarize the dynamics of both charges in the form of IPR. The red line indicates electrons, and the blue line indicates holes. Roman numbers in circles provided in panels (**d**,**e**) lable spatial positions of individual oligomers, as introduced in Figure 5.

**Table 1 polymers-16-01896-t001:** Absorption peaks for the single and ensemble of cis-PA on the basis of transitions between pairs of independent orbitals, and the dependence of rates on initial conditions. $E_{iE}-E_{LU}$ refers to the dissipated energy for the electron, and $E_{HO}-E_{iH}$ refers to the dissipated energy for the hole.

	Initial Transition	ΔE (eV)	$E_{iE}-E_{LU}\;(eV)$	$E_{HO}-E_{iH}\;(eV)$	$k_{e}\;(1/ps)$	$k_{h}\;(1/ps)$
A	HO→LU	1.1759	0	0	N/A	N/A
B	HO→LU+1	1.5299	0.3540	0	0.0434	N/A
C	HO−1→LU	1.5215	0	0.3456	N/A	0.0260
D	HO−1→LU+1	1.8755	0.3540	0.3456	0.0434	0.0260
C′	HO′→LU′+2	0.8122	0.1351	0	0.0408	0.0260
D′	HO′−2→LU′	0.8146	0	0.1375	N/A	0.0303
E	HO−2→LU+1	2.2983	0.3540	0.7684	0.0434	0.0303

**Table 2 polymers-16-01896-t002:** Relaxation rates of charge carriers for the single oligomer under different initial photo-excitations. $E_{iE}-E_{LU}$ refers to the dissipated energy for the electron, and $E_{HO}-E_{iH}$ refers to the dissipated energy for the hole.

No.	Initial Transition	$E_{iE}-E_{LU}\;(eV)$	$E_{HO}-E_{iH}\;(eV)$	$k_{e}\;(1/ps)$	$k_{h}\;(1/ps)$
1	HO−1 →LU+1	0.3540	0.3546	0.0434	0.0260
2	HO−2 →LU+2	0.7690	0.7684	0.0408	0.0339
3	HO−3 →LU+3	1.2017	1.2112	0.0323	0.0303
4	HO−1 →LU+2	0.7690	0.3546	0.0408	0.0260
5	HO−3 →LU+4	1.6364	1.2112	0.0303	0.0303
6	HO−4 →LU+3	1.2017	1.6457	0.0408	0.0273
7	HO−4 →LU+6	2.5320	1.6457	0.0289	0.0273
8	HO−3 →LU+2	0.7690	1.2112	0.0408	0.0303
9	HO−4 →LU+3	1.2017	1.6457	0.0323	0.0273
10	HO−1 →LU+3	1.2017	0.3546	0.0323	0.0260
11	HO−1→LU+4	1.6364	0.3546	0.0303	0.0260
12	HO−1 →LU+8	2.6454	0.3546	0.0312	0.0260
13	HO−4 →LU+1	0.3540	1.6457	0.0434	0.0273
14	HO−3 → LU+1	0.3540	1.2112	0.0434	0.0303
15	HO−4 → LU+7	2.5641	1.6457	0.0296	0.0273
16	HO−4 → LU+11	2.8152	1.6457	0.0313	0.0273

**Table 3 polymers-16-01896-t003:** Relaxation rates of charge carriers for the ensemble of oligomers under different initial photo-excitations. $E_{iE}-E_{LU}$ refers to the dissipated energy for the electron, and $E_{HO}-E_{iH}$ refers to the dissipated energy for the hole.

No.	Initial Transition	$E_{iE}-E_{LU}\;(eV)$	$E_{HO}-E_{iH}\;(eV)$	$k_{e}\;(1/ps)$	$k_{h}\;(1/ps)$
1	HO−6→LU+5	0.3048	0.5542	1.3975	0.5011
2	HO−6→LU+4	0.2500	0.5542	1.5594	0.5011
3	HO−5→LU+4	0.2500	0.3801	1.5594	1.0649
4	HO−4→LU+4	0.2500	0.3146	1.5594	0.9572
5	HO−6→LU+3	0.2048	0.5542	1.7015	0.5011
6	HO−5→LU+3	0.2048	0.3801	1.7015	1.0649
7	HO−4→LU+3	0.2048	0.3146	1.7015	0.9572
8	HO−3→LU+3	0.2048	0.3057	1.7015	1.3329
9	HO−6→LU+2	0.1351	0.5542	1.8881	0.5011
10	HO−5→LU+2	0.1351	0.3801	1.8881	1.0649
11	HO−4→LU+2	0.1351	0.3146	1.8881	0.9572
12	HO−3→LU+2	0.1351	0.3057	1.8881	1.3329
13	HO−6→LU+1	0.0925	0.5542	2.0278	0.5011
14	HO−5→LU+1	0.0925	0.3801	2.0278	1.0649
15	HO−4→LU+1	0.0925	0.3146	2.0278	0.9572
16	HO−3→LU+1	0.0925	0.3057	2.0278	1.3329

## Data Availability

Data are contained within the article and Appendix A.

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
