# Peer review of "Inter-Oligomer Interaction Influence on Photoluminescence in Cis-Polyacetylene Semiconductor Materials"

_polymers, 2024, doi:10.3390/polym16131896_

Round 1
Reviewer 1 Report
Comments and Suggestions for Authors
This manuscript reports on a theoretical study of the electronic and optical properties of cis-polyacetylene oligomers in the solid state. The computation methodology is appropriate but the work is marred by major flaws that must be corrected. Among those problems:
- The size of the periodic box (page 8) seems utterly arbitrary. This is a major issue because it has huge consequences on the conformational behavior of the molecular systems under study. In fact, it forces the molecules to be non-straight and non-planar, in contrast to previous studies (both experimental and theoretical) on cis-PA. This issue must be addressed in detail and fully justified.
- Along the same line, what is the distance between the oligomers in the ensemble calculations ? How was it chosen ? This is a central parameter because it governs the strength of the intermolecular interactions. This issue is not treated at all here (not even mentioned). This is a major weakness of this manuscript, which absolutely needs discussion and justification.
- The text on page 9 mentions a HOMO-LUMO gap of 1.2 eV for the single oligomer molecule. The value is smaller than that of the trans-polyacetylene polymer, which is very strange. Actually, the bandgap value is notoriously difficult to estimate accurately with DFT methods. Therefore the text must comprise a detailed discussion on the bandgap value calculated here.
- It is dismaying that nowhere in the manuscript the authors try to compare their results (eg, on the absorption and emission spectra) with experimental data. This is a huge weakness of this work and it must absolutely be corrected with a deep reworking of the text.
Minor points:
- The studies by Friend et al and Heeger et al deal with TRANS-polyacetylene and their findings (eg, the presence of solitons) do not apply to cis-polyacetylene, which is studied here. The corresponding paragraph in the introduction must therefore be corrected.
- Why are the molecules end-substituted by methyl groups and not simply with hydrogen atoms. This must be justified.
- The sentence describing the molecular orbitals (beginning of the results section) is impossible to understand. Why using two different notations for a given orbital ? This must be improved.
To conclude, my opinion is that this manuscript shows major weaknesses and it could be considered for publication only after a very deep and comprehensive revision.
Comments on the Quality of English Language
N/A
Author Response
REVIEWER #1:
Q1.1 This manuscript reports on a theoretical study of the electronic and optical properties of cis-polyacetylene oligomers in the solid state.
A1.1 Authors thank the Reviewer. Yes, this manuscript focuses on a theoretical investigation into the electronic and optical properties such as linewidth and lifetime of cis-polyacetylene oligomers in the solid state. Using advanced computational methods, we aim to provide a detailed understanding of how these properties influence potential applications in organic electronics, particularly exploring the effects of molecular structure on these properties. The following text has been added to
Page 15 lines 466-467
Adding additional defects, such as kinks, non-plater, twists would provide additional broadening.
A1.1, continued: The following text has been added to: Page 1 lines 24-26
This comparative study suggests the dispersion forces and orbital hybridizations between chains are the leading contributors to the variation in PL.
A1.1, continued: The following text has been added to: Page 15 lines 437-442
A systematic study of dependence on length of the chain is known in the literature and is going on as a continuation of this work. An increase of chain length is expected to provide a mild redshift of transition energies. Our exploration into models with shorter-length oligomers suggests a decrease in computed conductivity. This decrease is attributed to quantum confinement effects in shorter oligomers, which increase band gaps and sub-band gaps, subsequently reducing conductivity.
A1.1, continued: The following text has been added to: Page 15 lines 469-470
In the current study, dispersion forces and orbital hybridizations between chains are the major effect.
Q1.2 The computation methodology is appropriate, but the work is marred by major flaws that must be corrected. Among those problems:
- The size of the periodic box (page 8) seems utterly arbitrary.
A1.2 Authors thank the Reviewer. the following text has been added to Page 6, lines 200-201:
We explored the computational efficiency of forming periodic boxes filled with both shorter and longer oligomers, which could potentially be extended in future work.
A1.2 (continued) The following text has been added to Page 14, lines 447-455
In shorter oligomers, the confinement is more pronounced, affecting the electronic properties significantly. In contrast, longer oligomers exhibit transition energies that stabilize and are defined predominantly by the size of the exciton and polaron rather than by further increases in length. This behavior aligns with observations in the literature suggesting a general trend for exciton behavior in conjugated polymers. For instance, Kraner et al. noted that exciton sizes increase uniformly with molecular π-system sizes ranging from 10 to 40 Å 59 Beyond this range, no significant increase in exciton size was observed. These findings guided our decision to utilize shorter oligomers.
Q1.3 This is a major issue because it has huge consequences on the conformational behavior of the molecular systems under study. In fact, it forces the molecules to be non-straight and non-planar, in contrast to previous studies (both experimental and theoretical) on cis-PA. This issue must be addressed in detail and fully justified.
A1.3 Authors thank the Reviewer. Reviewer is correct. The following text has been added to Page 6 line 202-204
Changing the size of the box and practically thermal annealing would likely result in additional defects such as twists and kinks in individual oligomers. The current work corresponds to low temperature crystallization limit and can be considered as a reference point.
Q1.4- Along the same line, what is the distance between the oligomers in the ensemble calculations? How was it chosen? This is a central parameter because it governs the strength of the intermolecular interactions. This issue is not treated at all here (not even mentioned). This is a major weakness of this manuscript, which absolutely needs discussion and justification.
A1.4 Authors thank the Reviewer. the following text has been added to Page 6 line 196-199
For the ensemble model, oligomers are placed at closest possible distance to balance pressure between unit cells and achieve a realistic density. The optimal distance depends on dispersion corrections implemented by the D3 method56. The distance between nearest (5th and 6th) chains is 3.18 Å.
Q1.5- The text on page 9 mentions a HOMO-LUMO gap of 1.2 eV for the single oligomer molecule. The value is smaller than that of the trans-polyacetylene polymer, which is very strange.
A1.5 Authors thank the Reviewer. Please see both A1.5 and A1.6. The following text has been added to Page 2 line 66-73
Photogenerated excitations in trans-PA tend to relax into soliton states, significantly reducing PL in the visible range while primarily confining it to the infrared range. This characteristic, stemming from the unique topological and electronic structures of trans-PA, inherently limits its effectiveness in applications requiring visible-range PL. In contrast, cis-PA does not exhibit this soliton-induced reduction in PL, making it suitable for broader applications in optoelectronic devices where efficient and strong light emission is essential for novel applications in material science and electronics 22,24.
Q1.6 Actually, the bandgap value is notoriously difficult to estimate accurately with DFT methods. Therefore the text must comprise a detailed discussion on the bandgap value calculated here.
A1.6 Authors thank the Reviewer. The following text has been added to Page 7-8, lines 257-266:
In our earlier work 30, we did show that TDDFT calculations with a hybrid functional provide gap values comparable to experimental results. By comparing DFT and TDDFT data, one may extract a systematic correction. For a single oligomer, and .. The difference is . Thus, each transition energy in our calculations may be corrected as follows: An alternative way was to scale the energies by a factor of , where . In this approach, all transition energies in our calculations may be corrected as follows . 30
A1.6 (continued): The figures of Natural Transition Orbitals and TDDFT spectra form our previous work is included here as a reference. Please check Figure 3(from ref 15) and Table S1(from ref. 15) below.
Figure 3 (from https://doi.org/10.1080/00268976.2022.2110167 ). Experimental and calculated ground-state absorption spectra of the polyacetylene. (a) The absorption spectra were calculated using the PBE GGA function and the calculated absorption spectra using HSE06 hybrid functions [27]. (b) Energy-shifted absorption calculated using the PBE function and the experimental absorption of cis- and trans-poly(1-ethynyl-pyrene) molecules [28]. The labeling of the transitions is the same as in Figure 2.
Table S1 (from https://doi.org/10.1080/00268976.2022.2110167 ).: Hole and electron NTOs of the complex OLIG. All NTOs are Calculated using TDDFT on the ground state geometry with PBE functional and LANL2dz/6-31G* basis set with dichloromethane solvent using CPCM method.
|
OLIG |
Hole |
Electron |
|
S1 E = 1.90 eV f=3.7777 |
|
|
|
S4 E = 2.67 eV f=0.4562 |
|
|
|
S8 |
|
|
|
S10 E = 3.64 eV f=0.0227 H/L : 64% |
|
|
|
S11 E = 3.71 eV f=0.4216 |
|
|
|
S14 E = 3.88 eV f=0.1393 H/L : 39% |
|
|
|
S19 E = 4.41 eV f=0.0807 H/L : 33% |
|
|
|
S24 E = 4.79 eV f=0.2097 |
|
|
Q1.7- It is dismaying that nowhere in the manuscript the authors try to compare their results (eg, on the absorption and emission spectra) with experimental data.
A1.7 Authors thank the Reviewer. Please see A1.6 above. The following text has been added to Page 15 line 483-490
The current investigation is designed to deepen the theoretical understanding and enhance the accuracy of computational models within a defined theoretical framework. While experimental validation, partially provided in ref. 58,60 remains critical for the broader application of these computational study, the present study is limited to comparative analyses between different computational models. The computation-to-experiment comparison, while highly valuable, falls outside the scope of the current study. We expect that adding the scissor operator or scaling correction would lead to accurate transition energies and bring the spectra in computational results closer to experimental values.
Q1.8 This is a huge weakness of this work and it must absolutely be corrected with a deep reworking of the text.
A1.8 Authors thank the Reviewer. Please see A1.1. The following text has been added to Page 14 line 437-442
A systematic study of dependence on length of the chain is known in the literature and is going on as a continuation of this work. An increase of chain length is expected to provide a mild redshift of transition energies. Our exploration into models with shorter-length oligomers suggests a decrease in computed conductivity. This decrease is attributed to quantum confinement effects in shorter oligomers, which increase band gaps and sub-band gaps, subsequently reducing conductivity.
Q1.9 Minor points:- The studies by Friend et al and Heeger et al deal with TRANS-polyacetylene and their findings (eg, the presence of solitons) do not apply to cis-polyacetylene, which is studied here. The corresponding paragraph in the introduction must therefore be corrected.
A1.9 Authors thank the Reviewer. This para removed from introduction section.
Q1.10- Why are the molecules end-substituted by methyl groups and not simply with hydrogen atoms. This must be justified.
A1.10 Authors thank the Reviewer. the following text has been added to Page 15 line 471-482
The incorporation of methyl groups as terminal substituents in our oligomer configurations offers significant benefits. Methyl groups enhance molecular stability by providing steric protection, which reduces the susceptibility of the polymer ends to oxidative degradation. This protection is crucial for maintaining the structural integrity of the oligomers during experimental procedures and analyses. Additionally, these methyl groups significantly influence the electronic structures of the oligomers. They adjust the electron density along the polymer backbone, thereby affecting the bandgap and electronic transitions crucial for the PL properties under study. The strategic inclusion of methyl groups modifies the energy states of HO and LU, enhancing the optical properties of the oligomers. Therefore, the choice to use methyl groups instead of hydrogen atoms is not merely a structural decision but a strategic one aimed at optimizing the optoelectronic properties of the oligomers.
Q1.11- The sentence describing the molecular orbitals (beginning of the results section) is impossible to understand. Why using two different notations for a given orbital? This must be improved.
A1.11 Authors thank the Reviewer. the following text has been added to Page 6, lines 219-225:
. This notation is according to quantum confinement paradigm. A way to count orbitals of electrons upward from CBM and holes downward from VBM is an efficient way to establish an analogy with momentum dispersion in direct gap infinite polymers; ; . Note the negative size of dispersion for holes. For finite size oligomers both follow the trend of particle in a box, reflected in special pattern of subgaps. These background concepts provide a foundation for more organized analysis of electronic structures.
Q1.12 To conclude, my opinion is that this manuscript shows major weaknesses and it could be considered for publication only after a very deep and comprehensive revision.
A1.12 Authors thank the Reviewer. The constructive and inspiring suggestions have been taken into account! We addressed all reviewer comments and explained our work.

Reviewer 2 Report
Comments and Suggestions for Authors
1. The authors of the article carried out a colossal study, the purpose of which was to study the complex behavior of charge carriers, which make it possible to determine possible relaxation processes for both individual and oligomeric forms of cis-polyacetylene. The question arises: will the results obtained be reproducible for other classes of polymers/oligomers? What data can confirm this?
2. Figure 1 shows the structure of the oligomer cis-polyacetylene and an assembled group of six oligomer chains of cis-PA. The authors studied the processes of nonradiative relaxation of photoexcited electrons and holes for these structures. The polymer, polyacetylene, has an average molecular weight of 1000. The authors have a molecular weight for cis-polyacetylene that they calculated (C32H36) to be 420. The question arises: will the identified patterns be characteristic of a product with a molecular weight of about 1000?
3. One of the conclusions of the study was the necessary functionalization of conjugated polymers with side chains, such as alkyl groups, in order to isolate the oligomers from each other and, as a result, increase the fluorescence intensity. According to the authors of the article, these results are of significant importance for optimizing the molecular design of organic semiconductors for the purpose of their application in photovoltaic and LED technologies. However, it is worth noting that this fact has long been known and has been successfully implemented in numerous research articles. For example, an article was published on the role of alkyl side chains (Lei T., Wang J. Y., Pei J. Roles of flexible chains in organic semiconducting materials //Chemistry of Materials. – 2014. – Т. 26. – â„–. 1. – С. 594-603.).
However, the proposed modification of the conjugated polymer is only one of many principles that must be taken into account when developing conjugated polymers for specific applications. (Guo X., Baumgarten M., Müllen K. Designing π-conjugated polymers for organic electronics //Progress in Polymer Science. – 2013. – Т. 38. – â„–. 12. – С. 1832-1908.)
Therefore, the question remains: what contribution does the presented research make from a practical point of view in the field of semiconductor materials for organic electronics?
4. I would also like the authors to clearly formulate the scientific novelty of this study.
5. References must be formatted in accordance with the requirements of the journal
6. Remove frames for equations 13 a-d, 14
Author Response
REWIEVER 2
Q2.1 1. The authors of the article carried out a colossal study, the purpose of which was to study the complex behavior of charge carriers, which make it possible to determine possible relaxation processes for both individual and oligomeric forms of cis-polyacetylene.
A2.1 Authors thank the Reviewer. The following text has been added to Page 14-15 line 456-470
We focus on the influences and consequences of inter-chain interaction on nonradiative relaxation and spectral line width. The fundamental photophysical properties such as charge carrier dynamics and PL behavior are inherent to the conjugated polymer system of cis-PA. These properties demonstrated consistency across the different molecular weights we studied, suggesting that the trends we observed could extend to cis-PA products with broader molecular weights. As soon as the chain length exceeds the size of polaron, its influence on such observables becomes negligible. There is ongoing research to confirm this hypothesis. Adding additional defects, such as kinks, non-planar structures, twists will provide additional effects. This approach aligns with the comprehensive design strategies outlined in the literature, providing a foundation for our continued research into developing more efficient and application-specific CPs. In the current study, dispersion forces and orbital hybridizations between chains are the major effect.
Q2.2 The question arises: will the results obtained be reproducible for other classes of polymers/oligomers? What data can confirm this?
A2.2 Authors thank the Reviewer. the following text has been added to Page 15, line 467-469
This approach aligns with the comprehensive design strategies outlined in the literature, providing a foundation for our continued research into developing more efficient and application-specific CPs.
Q2.3 2. Figure 1 shows the structure of the oligomer cis-polyacetylene and an assembled group of six oligomer chains of cis-PA. The authors studied the processes of nonradiative relaxation of photoexcited electrons and holes for these structures.
A2.3 Authors thank the Reviewer for the summary!. Yes, in the manuscript we explained the nonradiative relaxation of photoexcited electrons and holes for single and ensemble oligomers of cis-PA.
Q2.4 The polymer, polyacetylene, has an average molecular weight of 1000. The authors have a molecular weight for cis-polyacetylene that they calculated (C32H36) to be 420. The question arises: will the identified patterns be characteristic of a product with a molecular weight of about 1000?
A2.4 Authors thank the Reviewer. We pursue this exploration and the convergence of gaps for longer chains is yet confirmend by prelimnary results, but the inter-oligomer hybridization overweights dependence on the chain length, it is as stronger effect. The following text has been added to Page 15 line 462-464
These properties demonstrated consistency across the different molecular weights we studied, suggesting that the trends we observed could extend to cis-PA products with broader molecular weights.
Q2.5 3. One of the conclusions of the study was the necessary functionalization of conjugated polymers with side chains, such as alkyl groups, in order to isolate the oligomers from each other and, as a result, increase the fluorescence intensity.
A2.5 Authors thank the Reviewer. the following text has been added to Pages 2, lines 56-60
The functionalization of CPs with flexible side chains, including alkyl groups, leads to enhancement of the solubility and device performance. These side chains do not directly participate in charge transport, but they can influence the overall device performance by affecting the molecular packing and physical robustness of the films 18.
Q2.6 According to the authors of the article, these results are of significant importance for optimizing the molecular design of organic semiconductors for the purpose of their application in photovoltaic and LED technologies. However, it is worth noting that this fact has long been known and has been successfully implemented in numerous research articles.
A2.6 Authors thank the Reviewer. We have reduced the degree of emphasis of this statement and acknowledged earlier experimental works! Please see Page 15 line 512-514
This work provides a detailed analysis of the optoelectronic properties of cis-PA, which could be used for improving nanostructured semiconductor materials for photovoltaic and film-forming conductive polymers processing.
Q2.7 For example, an article was published on the role of alkyl side chains (Lei T., Wang J. Y., Pei J. Roles of flexible chains in organic semiconducting materials //Chemistry of Materials. – 2014. – Т. 26. – â„–. 1. – С. 594-603.).
A2.7 Authors thank the Reviewer. Answered in A2.5, please see Pages 2, lines 56-60
Q2.8 However, the proposed modification of the conjugated polymer is only one of many principles that must be taken into account when developing conjugated polymers for specific applications. (Guo X., Baumgarten M., Müllen K. Designing π-conjugated polymers for organic electronics //Progress in Polymer Science. – 2013. – Т. 38. – â„–. 12. – С. 1832-1908.)
A2.8 Authors thank the Reviewer. the following text has been added to Page 15 line 483-487
The current investigation is designed to deepen the theoretical understanding and enhance the accuracy of computational models within a defined theoretical framework. While experimental validation, partially provided in ref. 60 remains critical for the broader application of these computational study, the present study is limited to comparative analyses between different computational models.
(60) Guo, X.; Baumgarten, M.; Müllen, K. Designing π-Conjugated Polymers for Organic Electronics. Prog Polym Sci 2013, 38 (12), 1832–1908. https://doi.org/10.1016/j.progpolymsci.2013.09.005.
Q2.9 Therefore, the question remains: what contribution does the presented research make from a practical point of view in the field of semiconductor materials for organic electronics?
A2.9 Authors thank the Reviewer. regarding the practical contributions of our research in the field of semiconductor materials for organic electronics. Page 15 line 497-500
We report novel insights into nonadiabatic dynamics of photoexcited electrons and holes, showcasing distinct relaxation rates under isolated versus ensemble conditions. Our findings quantify relaxation timescales, enhancing the design and efficiency of photovoltaic materials..
For a detailed explanation of these contributions, please refer to the specified sections in the manuscript which provide a thorough discussion of how our research contributes to advancing the field of organic electronics and enhancing device functionality.
Q2.10 4. I would also like the authors to clearly formulate the scientific novelty of this study.
A2.10 We appreciate your request for clarification on the scientific novelty of our research. The novel contributions and implications of our study are comprehensively discussed in the conclusion section of our manuscript, We believe this section thoroughly addresses the unique aspects of our findings and their relevance to the field of conjugated polymers. Thank you for the opportunity to highlight this part of our work. Please see specifically on Pages 15 lines 502-503.
Our comparative study suggests the dispersion forces and orbital hybridizations between chains are the leading contributors to the variation in PL. We thus expect that functionalization of CPs by side chains will mitigate inter-oligomer interactions. This approach is anticipated to enhance the intensity and stability of PL..
Q2.11 5. References must be formatted in accordance with the requirements of the journal
A2.11 Authors thank the Reviewer. References have been reformatted.
Q2.12 6. Remove frames for equations 13 a-d, 14
A2.12 Authors thank the Reviewer. Frame already removed from all equations.

Round 2
Reviewer 1 Report
Comments and Suggestions for Authors
The authors have properly addressed most of my comments/suggestions and the manuscript can now be accepted for publication.
Comments on the Quality of English LanguageN/A
Author Response
Reviewer 1: Comments and Suggestions for Authors
Q1.1 The authors have properly addressed most of my comments/suggestions and the manuscript can now be accepted for publication.
A1.1 Authors are grateful to Reviewer 1 for an effort of evaluating our work and a positive review!
Updated version of the manuscript went through another iteration of proofreading, included minor English editing as suggested!
Reviewer 2 Report
Comments and Suggestions for Authors
We would like to thank the authors for making significant changes to the article that have improved its quality. In this regard, the presented material may be recommended for publication in the journal «Polymers».
However, unfortunately, the remark regarding the design of the list of references remained uncorrected. Namely, the instructions for authors indicate that the names of journals should be indicated as their abbreviation. However, references 1, 2, 5, 10, etc. contain the full name of the journals. Therefore, we recommend that authors once again read the instructions for authors and make changes to the design of the references.
Author Response
Comments from Reviewer #2
Q2.1. We would like to thank the authors for making significant changes to the article that have improved its quality. In this regard, the presented material may be recommended for publication in the journal «Polymers».
A2.1 Authors thank Reviewer #2 for investing precious time into evaluation and positive recommendation of our work!
Q2.2 However, unfortunately, the remark regarding the design of the list of references remained uncorrected. Namely, the instructions for authors indicate that the names of journals should be indicated as their abbreviation. However, references 1, 2, 5, 10, etc. contain the full name of the journals. Therefore, we recommend that authors once again read the instructions for authors and make changes to the design of the references.
A2.2 Authors thank Reviewer #2 for careful attention! All references were doublechecked to match the right format. the references mantions by the Reviewer are quoted below in the updated format.
(1) Xu, Y.; Cui, Y.; Yao, H.; Zhang, T.; Zhang, J.; Ma, L.; Wang, J.; Wei, Z.; Hou, J. A New Conjugated Polymer That Enables the Integration of Photovoltaic and Light-Emitting Functions in One Device. Adv. Mater. 2021, 33 (22), 2101090. https://doi.org/10.1002/adma.202101090.
(2) Liu, Y.; Yan, S.; Ren, Z. π-Conjugated Polymeric Light Emitting Diodes with Sky-Blue Emission by Employing Thermally Activated Delayed Fluorescence Mechanism. Chem. Eng. J. 2021, 417, 128089. https://doi.org/10.1016/j.cej.2020.128089.
(5) Yıldız, D. E.; Cevher, D.; Yasa, M.; Cirpan, A.; Toppare, L. Selenophene-Containing Conjugated Polymers for Supercapacitor Electrodes. J. Polym. Sci. 2022, 60 (1), 109–121. https://doi.org/10.1002/pol.20210746.
(10) Garg, S.; Goel, N. Optoelectronic Applications of Conjugated Organic Polymers: Influence of Donor/Acceptor Groups through Density Functional Studies. J. Phys. Chem. C 2022, 126 (22), 9313–9323. https://doi.org/10.1021/acs.jpcc.2c02938.